# Dynamic Behaviors of Mortar Reinforced with NiTi SMA Fibers under Impact Compressive Loading

**DOI:** 10.3390/ma14174933

**Published:** 2021-08-30

**Authors:** Eusoo Choi, Ha-Vinh Ho, Junwon Seo

**Affiliations:** 1Department of Civil and Environmental Engineering, Hongik University, Seoul 04066, Korea; hovinhha@mail.hongik.ac.kr; 2Department of Civil and Environmental Engineering, South Dakota State University, Brookings, SD 57007, USA; junwon.seo@sdstate.edu

**Keywords:** split Hopkinson pressure bar (SHPB), SMA fibers, impact compressive loading, energy absorption

## Abstract

In this study, a compressive impact test was conducted using the split Hopkinson pressure bar (SHPB) method to investigate SMA fiber-reinforced mortar’s impact behavior. A 1.5% fiber volume of crimped fibers and dog-bone-shaped fibers was used, and half of the specimens were heated to induce recovery stress. The results showed that the appearance of SMA fibers, recovery stress, and composite capacity can increase strain rate. For mechanical properties, the SMA fibers reduced dynamic compressive strength and increased the peak strain. The specific energy absorption of the reinforced specimens slightly increased due to the addition of SMA fibers and the recovery stress; however, the effect was not significant. The composite behavior between SMA fibers and the mortar matrix, however, significantly influenced the dynamic compressive properties. The higher composite capacity of the SMA fibers produced lower dynamic compressive strength, higher peak strain, and higher specific energy absorption. The composite behavior of the dog-bone-shaped fiber was less than that of the crimped fiber and was reduced due to heating, while that of the crimped fiber was not. The mechanical properties of the impacted specimen followed a linear function of strain rate ranging from 10 to 17 s^−1^; at the higher strain rates of about 49–67 s^−1^, the linear functions disappeared. The elastic modulus of the specimen was independent of the strain rate, but it was dependent on the correlation between the elastic moduli of the SMA fibers and the mortar matrix.

## 1. Introduction

Structures made from cementitious material are subjected to various dynamic loadings such as impact loading, explosion, or penetration. Thus, a better understanding of their dynamic properties is necessary; the mechanical behavior of concrete/mortar under dynamic loading is significantly different from that of concrete/mortar subjected to static loading [1,2]. Dynamic tests are more complex than static tests with more parameters, such as inertia effect, specimen geometry, and stress-wave propagation, affecting the experimental results. Various testing methods have achieved a wide range of strain rates from 10^−5^ s^−1^ for the static case to 10^2^ s^−1^ for high-level strain rate [3].

For concrete/mortar, the drop weight method with strain rates ranging from 1 to 10 s^−1^ is commonly used because it is easy to conduct under normal conditions. This method is passive because the impact loading cannot be controlled for a higher strain rate; moreover, the results are greatly affected by specimen size and configuration, such as drop weight, speed, and support stiffness [4,5]. Thus, the drop weight method may not be suitable in some cases. The split Hopkinson pressure bar (SHPB) technique is commonly used to characterize the dynamic compressive behavior of materials with strain rates ranging from 10 to 10^3^ s^−1^. This method has been employed to investigate the dynamic behavior of mortar [6], normal concrete [7], high-strength concrete [8], fiber-reinforced concrete [9], and so on.

The mechanical properties of concrete/mortar, such as strength, stiffness, and brittleness (or ductility), are greatly influenced by the strain rate. For example, the compressive strength increases with the strain rate; the increasing trend is dramatic when the strain rate is greater than 30 s^−1^ [10]. The strain rate affects the compressive strength of concrete/mortar for two reasons: the time-dependent microcrack growth and the viscoelastic character of the hardened cement paste [11]. The dynamic increase factor (DIF), which is defined as the ratio between dynamic and quasi-static strength of the specimen, is widely used to illustrate the strain rate effects on mortar or concrete [9,12]. The DIF is used for both the analysis and design of concrete structures. Various DIFs have been suggested and presented in codes ACI 349-13 [13], ACI 370R-14 [14], and UFC 3-340-02 [15].

Fibers are embedded in mortar/concrete to increase their strength, toughness, and resistance to impact [16]. The dynamic properties of reinforced concrete/mortar depend on the type of fiber and fiber content. Xu et al. [4] added steel and synthetic fibers to concrete and found that the rate sensitivity and mechanical properties of reinforced concrete under compressive impact loading are dependent on the material properties of the embedded fiber and the fiber’s shape. Better anchoring bonds and mechanical deformation of the steel fiber cause the higher rate sensitivity of toughness. Various studies have indicated that the increase of the toughness of concrete at high strain rates can be attributed to the higher strength and strains [17,18]. Ren et al. [17] indicated that the steel fibers have little effect on elastic modulus and peak strain while they slightly increased dynamic compressive strength.

Shape memory alloy (SMA) materials were invented some decades ago, and SMA fibers have recently been researched as reinforcements to cementitious materials. Because of the unique properties of the shape memory effect and super-elasticity, SMA fibers have some advantages in crack-closing, pre-stressing, crack-repairing, and self-centering [19,20]. Many SMA fiber shapes, such as dog-boned-shaped, L-shaped, N-shaped, paddle-shaped, and crimped, have been introduced [21]. Crimped and dog-boned-shaped fibers have shown great potential because they have high pullout resistance and are mass-produced [22,23,24]. A dog-bone-shaped fiber is easily created by heating the end parts, while a crimped fiber is produced continuously by a rolling device. Thus, in previous studies we mixed the two fibers mentioned in the mortar matrix to investigate the reinforced mortar’s uniaxial monotonic and cyclic compressive behavior [24,25]. The fibers were verified to positively improve the mechanical properties of reinforced mortar under various compressive loads. For example, the fibers increased the compressive strength in a monotonic test and reduced the plastic strain in a cyclic test. Moreover, the higher fiber content and recovery stress in the case of heating are two main factors that significantly improve the characteristics of reinforced mortar.

The SMA fiber with the shape memory effect provides a passive bond resistance as well as an active one. The passive bond resistance is induced mainly by a mechanical bond, which depends on the fiber’s shape and mechanical properties; steel fibers and other types of fiber provide only passive bond resistance. However, the active bond resistance of the SMA fiber is induced by the recovery stress due to the shape memory effect; thus, the recovery stress provides compressive prestress in the mortar before the application of loading. It was found that that the recovery stress values of the crimped and dog-bone shaped SMA fibers were positive, and they increased the compressive strength of mortar under static loading conditions. Thus, the effect of recovery stress as an active bond resistance on mortar under dynamic or impact loading conditions should be investigated. This study aimed to investigate the dynamic compressive behavior of reinforced mortar using the two types of SMA fibers, crimped and dog-bone-shaped fibers, under impact loading. The split Hopkinson pressure bar (SHPB) test was used to apply impact loading. The dynamic compressive strength, peak strain, elastic modulus, and energy absorption of the plain and reinforced specimens were examined.

## 2. Experimental Test

### 2.1. Specimens

The Ni50.4-Ti (wt.) SMA fibers were used in this study. Both crimped and dog-bone-shaped fibers had the same length of 30 mm and an initial diameter of 0.956 mm. The dimensions of SMA fibers are presented in Figure 1, whereas the direct tensile behavior and pullout behavior are shown in Figure 2 and Figure 3, respectively. The production of the SMA fibers and details of the tensile test and pullout test were presented in recent studies [22,23,24,25,26]. The crimped and dog-bone-shaped fibers produce the same tensile behavior in heating and non-heating with the yield stress of 950 MPa. Crimped fibers are slightly different from straight fibers with a 0.073 mm wave depth of indentations; however, they produce a relatively high pullout resistance of about 700 MPa. This wave depth guarantees that the SMA fibers contribute fully to the reinforced mortar characteristics without being yielded to or ruptured. Dog-bone-shaped fibers also significantly increase the pullout resistance because of the larger bulged diameter in the end parts. However, the end-anchoring resistance of the dog-bone-shaped fiber seems to be less effective than the multiple crimped anchoring of the crimped fiber. The maximum resistance of dog-bone-shaped fibers in non-heating and heating is 80 MPa and 200 MPa, respectively. Table 1 shows the mixed proportions of plain mortar. ASTM type I normal Portland cement, fly ash, silica sand, and water-reducing admixture were used for all mortar mixtures. The specific gravity of sand was 1.5, and the maximum size was 0.4 mm. For reinforced mortar, 1.5% fiber content of crimped or dog-bone-shaped SMA fibers was added to the mixture.

The specimen dimensions were chosen to evaluate the effect of specimen size on heterogeneity. The specimen diameter should be at least three times the nominal maximal coarse aggregate size, while the thickness should not be larger than the diameter [5,27]. ASTM C192/C192M-16a [28] contains the following recommendations for the making and curing of a specimen. First, a D75 × H150 mm cylinder mold is used to cast the specimens. After 28 days, the D75 × H75 mm specimens, which are the middle-half parts of the D75 × H150 mm specimens, are made by cutting both ends away. Half of the specimens are heated in an oven with a constant temperature of 150 °C to introduce the recovery stress of the embedded SMA fibers. The 150 °C is higher than the finishing austenite phase temperature of 93.3 °C, which is suggested by the differential scanning calorimetry test of the manufacturer; thus, the SMA fibers perfectly complete phase transformation and produce the recovery stress. Moreover, that temperature does not harm the mortar’s properties. In recent studies the oven heating method was used to investigate the effect of recovery stress on the reinforced mortar’s monotonic, cyclic compressive behavior [24,25]. The reinforced specimens, made with either crimped or dog-bone-shaped SMA fibers, were named CR specimens and DG specimens, respectively, whereas the plain specimen was called P specimen. The letter “H” denotes that the specimen is heated before the SHPB test is conducted. Six types of specimens [P, CR, DG, P(H), CR(H), DG(H)] were prepared, as shown in Figure 4.

### 2.2. SHPB Test

Figure 5 shows a schematic of the SHPB test. In an SHPB test, the air gun pushes the steel striker bar, which then impacts the steel incident bar. Thus, a dynamic compressive wave will propagate in the steel incident bar (incident wave). When the incident wave reaches the steel incident bar and specimen interface, a reflected wave is created due to the mismatching impedance between the bar and specimen, and another transmitted wave passes through the specimen. The transmitted wave is partly reflected and transmitted again at the interface between the specimen and the transmitted bar. The specimen’s stress and strain are measured using strain gauges attached to the incident and transmitted bars. The sampling rate of response is 2 MHz.

Each type of specimen was tested with impact velocities of 8 and 12 m/s. The notations V8 and V12 are used after the names of the specimens to indicate the applied impact velocity. The impact velocities were chosen to generate relatively moderate and high strain rates in mortar specimens. Each specimen was positioned between the incident and transmitted bars. Then, the impact loading fractured the specimen due to dilation in the radial direction (Figure 6). A high-speed camera recorded each specimen’s deformation process at a rate of 40,000 frames per second.

The dynamic compressive strength was calculated by averaging the front and back stresses; the front stress was at the interface between the incident bar and the specimen, whereas the back stress was at the interface between the specimen and the transmitted bar. A smaller difference between the front and back stresses means that the data are closer to the dynamic stress equilibrium state; the stresses are reliable. However, a high rate of increase of the incident stress wave can cause the premature failure of brittle material, such as mortar or reinforced mortar; thus, to slow down the increasing incident stress wave, a pulse shaper is widely used [7,27]. This study introduced an annular pulse shaper made of copper, which was attached to the impacted end of the incident bar (Figure 7). The dimensions of the pulse shaper were as follows: 52 mm outer diameter, 48 mm inner diameter, and 3 mm thickness. The pulse shaper and specimen were lubricated by petroleum jelly to minimize the frictional effect.

### 2.3. Data Analysis

The schematic of the strain pulse is shown in Figure 8. The specimen and the bars have the same cross-sectional area $\left(A_{s}\right)$. The specimen’s dynamic stress-strain relationship is based on the one-dimensional stress wave theory, and it is presented by Equations (1)–(5):(1)$$\sigma_{front}=\sigma_{i}+\sigma_{r}=E\left(\varepsilon_{i}+\varepsilon_{r}\right)$$
(2)$$\sigma_{back}=\sigma_{t}=E\varepsilon_{t}$$
(3)$$\sigma_{s}=\frac{E\left(\varepsilon_{i}+\varepsilon_{r}+\varepsilon_{t}\right)}{2}$$
(4)$$\dot{\varepsilon }=\frac{C_{v}}{l_{s}}(\varepsilon_{i}-\varepsilon_{r}-\varepsilon_{t})$$
(5)$$\varepsilon_{s}=\frac{C_{v}}{l_{s}}\int_{0}^{t}(\varepsilon_{i}-\varepsilon_{r}-\varepsilon_{t})dt$$
where $\sigma_{front},\;\sigma_{back},\;\sigma_{s},\;\dot{\varepsilon }$, and $\varepsilon_{s}$ are the front, back, and average stresses, the strain rate, and the strain of specimens, respectively. Here $\sigma_{i},\;\sigma_{r},\sigma_{t},\varepsilon_{i},\;\varepsilon_{r}$, and $\varepsilon_{t}$ denote the incident, reflected, and transmitted wave stresses, and the corresponding wave strains, respectively; E and $l_{s}$ are the elastic modulus and the specimen’s initial length, respectively, and $C_{v}$ is the wave velocity in bars.

The strain rate is adopted when the dynamic stress reaches the maximal value, as seen in Figure 9. In the figure, the y-left-axis denotes the dynamic compressive stress, while the y-right-axis is the strain rate, and the x-axis refers to the strain. The strain rate ranges from 10 to 17 s^−1^ with an impact velocity of 8 m/s and 49 to 67 s^−1^ with an impact velocity of 12 m/s (Table 2). The appearance of SMA fibers increases the strain rate of the reinforced specimens because SMA fibers are more homogeneous than is the mortar matrix, leading to more effective transfer impact waves. However, the increment is not significant: about 7 s^−1^ for an impact velocity of 8 m/s and 18 s^−1^ for an impact velocity of 12 m/s. The strain rate of the reinforced specimens is influenced by heat treatment, with the maximum difference in strain rate due to heating being 12 s^−1^ at an impact velocity of 12 m/s; however, the plain specimen is not influenced. Thus, the recovery stress in heated SMA fibers affected the transferred impact waves, resulting in a changing strain rate.

The shape of SMA fibers also affected strain rate. For non-heating specimens, crimped fibers are less effective in increasing the strain rate than are dog-bone-shaped fibers. However, the heated crimped fibers increase the strain rate, while the heated dog-bone-shaped fibers produce the opposite trend. This observation can be explained by a higher composite between dog-bone-shaped fibers and the mortar matrix due to the initial rigid end-anchoring resistance of the fiber in non-heating. However, when heated, the dog-bone-shaped fibers bulged in diameter along the whole length; thus, the frictional resistance increased, and the end-anchoring bond was lost. In contrast, the bond behavior of crimped fibers does not change much due to heating (Figure 3). Some recent studies investigated the bond behavior of crimped and dog-bone-shaped fibers on composite capacity in a mortar matrix [25].

R-value, which is the difference between front stress and back stress, evaluates a specimen’s dynamic stress equilibrium state. This value was first introduced by Flores-Johnson [27,29]. R-value is calculated as:(6)$$R=\left|\frac{\sigma_{front}-\sigma_{back}}{\sigma_{average}}\right|\times 100\%$$
where $\sigma_{front},\;\sigma_{back},\;\sigma_{average}$ are the front, back, and average stresses of the specimen, respectively. When the difference between the front and back stresses is not significant, the specimens are close to the dynamic equilibrium state. In a previous study with both types of D50 × H50 mm and D75 × H75 mm specimens, the dynamic equilibrium was satisfied when the R value is not higher than 10% [27]. As seen in Table 2 and Figure 10, R-values were less than 6%; only two specimens had 4% and 6% R-values, while the others had R-values ranging from 0% to 3%. Thus, the data from the experimental test are reliable.

## 3. Results and Discussion

### 3.1. Failure Mode

Figure 11 and Figure 12 show the failure modes of the specimens. In general, the failure mode changes from longitudinal splitting failure to pulverization failure with increasing strain rate. The specimens show several longitudinal cracks at a strain rate of less than 17 s^−1^, and they are crushed into small fragments with a strain rate higher than 49 s^−1^. A specimen is loaded rapidly in the longitudinal direction of loading at a high strain rate, but it does not expand much in the radial direction because of the inertial effect. Inertia restraint creates lateral inertia confinement, which prevents larger cracks and stimulates the appearance of cracks [8,9]. Thus, a high strain rate leads to an increasing number of cracks.

The influence of embedded fibers on the failure mode is observed clearly at a strain rate over 49 s^−1^. The SMA fibers reinforcement effectively transfers internal impact waves; thus, more elements in reinforced specimens undergo nearly equal stress. Moreover, the reinforced specimens are fractured into many fragments, but they are still attached to the fibers. Due to the fibers’ holding effect, the cracks continuously propagated in the fragments, resulting in smaller fragments than those in the plain specimen.

### 3.2. Dynamic Compressive stress-strain Curve

The relationships of dynamic compressive stress and strain of specimens are presented in Figure 13. The specific energy absorption is defined as the energy absorption per unit volume of material [9]. It is calculated as the area under the dynamic stress-strain curve. Based on the dynamic stress-strain curves, the specimens’ mechanical properties are presented in Table 3.

For the impact velocity of 8 m/s, the stress-strain response obtained has an obvious unloading phase; that means both stress and strain decrease with time after the peak stress. The unloading phase is essentially caused by the release of elastic energy stored in the specimen, which means that the specimen still has the load-bearing capacity, and the specimen can probably be compressed to higher stress and absorb more energy. In contrast, the impact at 12 m/s produces the typical dynamic stress-strain response, which is almost linear up to the peak stress and then shows a softening descending branch.

The curves of reinforced specimens are similar to those of plain specimens in both the non-heating and heating cases. However, dynamic compressive strength values are lower; this phenomenon was also observed in a previous study with steel fiber-reinforced concrete [30]. The reducing dynamic compressive strength of reinforced specimens is due to the higher porosity in the interfacial transition zone and the higher overlap of the interfacial transition zone aureoles. It is recognized that the dynamic strength, peak strain, and specific energy absorption of the specimens are sensitive to the strain rate in both cases of heat treatment. The same conclusions were drawn in previous studies with strain rates ranging from 40 to 200 s^−1^ [31,32,33].

### 3.3. Dynamic Compressive Strength

In general, the specimens showed increased dynamic compressive strength with increasing strain rate. Mortar/concrete is well-known to be sensitive to the loading rate; when the strain rate increased, the wave transferred rapidly through the aggregates and cement paste rather than around them [34]. Since the aggregates and cement paste are stronger than the interface transition zone, the dynamic compressive strength increased.

The dynamic compressive strengths of reinforced specimens are lower than those of plain specimens. As previously mentioned in the dynamic stress-strain curve section, the higher porosity and overlap of interfacial transition zone aureoles and the bridging effect of SMA fibers are the reasons for that. Moreover, the fibers on the crack paths slowed down the crack velocity due to the high pullout resistance and the fibers’ high tensile strength. It is well-known that increasing crack velocity increases dynamic compressive strength [32,34]. Thus, the slower crack velocity in reinforced specimens causes a lower increase in dynamic compressive strength.

At an impact velocity of 8 m/s, the dynamic compressive strength is a linear function of strain rate (Figure 14a). The dog-bone-shaped fibers produce more reduction than do the crimped fibers, and heating also reduces the dynamic compressive strength. At an impact velocity of 12 m/s, excluding the DG(H)_V12 specimen, the dynamic compressive strength reduces due to heating (Figure 14b). The reduction of the reinforced specimen is not different from that of the plain specimen; about 6% and 9%, respectively. Thus, the reduction of dynamic compressive strength is due to the damage of the mortar matrix caused by heating. The recovery stress of the heated SMA fibers does not have any effect on the increasing dynamic compressive behavior.

For DG(H) specimen, due to the low composite capacity of heated dog-bone-shaped fibers, the cracks easily propagate and rapidly develop. The reduced composite capacity of dog-bone-shaped fibers due to heating is explained in a previous study [25]. Dog-bone-shaped fiber is produced by heating both end parts of the fibers; thus, the fiber has anchorages of bulged diameters at both ends. When the specimen with dog-bone-shaped fibers is heated, DG(H) specimen, the embedded dog-bone-shaped fibers bulge along the whole length. Thus, the frictional resistance between the fiber and mortar matrix increases; however, the anchoring bond due of bulging in the diameter of the fiber ends is lost, which leads to the reduction of composite capacity of embedded dog-bone-shaped fibers and mortar when the reinforced specimen is heated. In contrast, the crimped fiber bulges in diameter and the anchoring bond due to its shape remains; thus, the heated crimped fiber is more adhered to the mortar matrix. When the strain rate is less than the transition strain rate of 30 s^−1^, the dynamic compressive strength is not influenced much; thus, the effect of composite behavior on dynamic compressive strength was not presented; however, the effect becomes significant with the strain rate higher than 30 s^−1^, resulting in higher compressive strength of the DG(H)_V12 specimen. The composite between dog-bone-shaped fibers and mortar matrix in DG(H)_V12 specimen reduces but is not lost totally. The heated dog-bone-shaped fibers still provide their bridging cracks because the dynamic compressive strength of the DG(H)_V12 specimen is still less than that of the P(H)_V12 specimen.

### 3.4. Peak Strain

In contrast with the dynamic compressive strength, the peak strain increases with increasing strain rate (Table 3). The propagation of stress waves at a high strain rate is more rapid than that of the cracks. Thus, the crack path is not around the mortar matrix and aggregates; rather, it is through them. As a result, the crack length is shorter, and the number of cracks increases. More cracks form before the peak stress, which are attributable to the increased peak strain [32]. The lateral confinement due to the inertial effect also contributes to the increased peak strain because it reduces the width of macro cracks in the direction perpendicular to the loading direction [5]. The SMA fibers produce higher transferring stress waves in the reinforced specimen; thus, more cracks appear, and as a result, the peak strains of the reinforced specimens are higher than that of the plain specimen. The dog-bone-shaped fibers produce a higher peak strain due to the lower composite capacity compared to that of the crimped fibers.

Heating causes the loss of pore water and harms the mortar’s properties, leading to more cracks; thus, the specimen’s peak strain increases with heating. However, the effect of heating on the specimen reinforced with dog-bone-shaped fibers is not significant because the dog-bone-shaped fibers reduce composite capacity when heated (Figure 15). The peak strain of the DG(H)_V8 specimen nearly equals that of the DG_V8 specimen, whereas at an impact velocity of 12 m/s, the peak strain of the heated specimen using dog-bone-shaped fibers slightly decreases. The peak strains of all specimens are a linear function of strain rate at an impact velocity of 8 m/s.

### 3.5. Energy Absorption Property

The specific energy absorption is about 250 kJ·m^−3^ at an impact velocity of 8 m/s and about 1150 kJ·m^−3^ at an impact velocity of 12 m/s; the specific energy absorption increases with increasing strain rate. This increasing trend is also observed in plain concrete and steel fiber-reinforced concrete [18].

At the same impact velocity, the specific energy absorption of the specimen increases when adding SMA fibers or/and heating (Figure 16). The increase in specific energy absorption due to SMA fibers is not large. The study by Jiao et al. [18] on steel fiber reached the same conclusion, with strain rates ranging from 30 to 100 s^−1^. The reason is that microcracks are initiated and propagated during impact loading, which requires energy. The fibers form a network that bridges the microcracks; thus, more energy is needed for microcrack development. Similar to the peak strain, the specific energy absorption is a linear function of strain rate at the impact velocity of 8 m/s.

### 3.6. Elastic Modulus

The strain rate did not affect the elastic modulus of plain and reinforced specimens. This behavior is different from that of concrete in a previous study [8]. For concrete, the strain velocity is much lower than the stress velocity due to the appearance of coarse aggregates such as gravel or limestone with large size and high elastic modulus. The strain wave takes a finite time to form around the coarse aggregates. The delayed response of the strain compared with the stress wave causes a decrease in the strain at a given stress. Thus, the elastic modulus of concrete increases with the strain rate. For mortar, sand with small particles does not reduce the strain velocity much. Therefore, the elastic modulus of mortar specimens does not change.

The elastic modulus of the specimens is presented in Figure 17, which indicates that the SMA fibers and heat treatment reduce the elastic modulus of mortar. According to Figure 2, the tensile stress-strain behavior of SMA fibers, the SMA fiber’s elastic modulus was about 22 GPa, which was lower than that of plain mortar with 24.1 GPa. Moreover, embedded fibers cause porosity in the reinforced mortar. Thus, the elastic modulus of the composite decreases. Previous studies have also observed this phenomenon with the low elastic modulus of synthetic fibers [35]. In a recent study of monotonic and cyclic compressive load, the elastic modulus of the reinforced specimen increases when adding a higher elastic modulus of SMA fibers into a mortar matrix with a lower elastic modulus. However, when the elastic modulus of the SMA fibers is less than that of the mortar matrix, the appearance of SMA fibers increases the elastic modulus of the reinforced specimen [25]. The heating process makes the mortar experience physical and chemical changes, such as evaporation of pore water, disintegration of aggregates, coarsening of microstructure, and increasing or porosity. The changes cause the deterioration of the mechanical properties of mortar [36,37]. Ma et al. [36] indicated that at below 400 °C, the primary mechanism for the decreasing elastic modulus is the vapour pressure caused by the evaporating pore water in the capillary pores. Thus, in this study, by heating at 150 °C, the elastic moduli of the specimens decreased.

## 4. Discussion of Results

The strain rates for the impact velocity of 8.0 m/s ranged from 10 to 17 s^−1^, and those for the 12.0 m/s were distributed from 49 and 67 s^−1^. Thus, as expected, the 8.0 m/s impact velocity generated a moderate strain rate, and the strain rate for the 12.0 m/s impact velocity was relatively high [8,38]; moderate ranges are near 10 s^−1^ and high ranges are from 40 to 300 s^−1^. For the plain mortar specimen under static loadings, heating at 150 °C did not decrease its compressive strength. For example, in the monotonic static compressive test, the heated specimen showed an increase in compressive strength of 11.9%, and with the cyclic static compressive test, the heated specimen showed the same compressive strength as that of the non-heated specimens [24]. Thus, the heating may induce micro-damage in the mortar, but the damage did not impute a negative effect or cause a decrease in the compressive strength. However, for the dynamic compressive test, the dynamic compressive strength of the mortar decreased by 1.1% with a strain rate of 11 s^−1^ and by 5.7% with a strain rate of 50 s^−1^. The decrement of dynamic compressive strength became larger with higher strain rates. Thus, it was conjectured that the dynamic compressive strength of the mortar is sensitive to microdamage, which is the outcome of heating. The impact with higher strain rates could reduce the compressive strength.

In a previous study that conducted static compressive tests of fiber-reinforced mortar with the CR and DG SMA fibers, their static compressive strengths increased compared to that of the plain unreinforced mortar. Moreover, the heated reinforced specimens showed higher compressive strength than those of the non-heated specimens and the heated plain specimen. Thus, the recovery stress of the SMA fiber induced by heating was assumed to act positively to increase compressive strength. However, unexpectedly, the dynamic compressive strengths of the mortar reinforced with the same types of SMA fibers decreased, and heating decreased the dynamic compressive strength of the reinforced specimens even further. In general, fiber reinforced mortar/concrete shows a relatively small dynamic compressive strength compared to that of plain unreinforced mortar/concrete. Glass fiber, polypropylene fiber, and carbon fiber showed negative effects on the dynamic compressive strength of mortar [38]. Steel fiber also decreased the dynamic compressive strength of high-strength concrete [8]. Thus, the decrement of the dynamic compressive strength of mortar reinforced with the SMA fibers is reasonable. However, the recovery stress induced by heating generated a negative effect on the dynamic compressive strength of the reinforced mortar in contrast to the static tests’ results. The DG(H)_V12 specimen showed higher dynamic compressive strength than that of the corresponding non-heated specimen, DG_V12. In that case, it was conjectured that the DG_V12 specimen showed a relatively low value because the two heated reinforced CR(H)_V12 and DG(H)_V12 specimens showed similar dynamic compressive strength. Thus, it is concluded that the recovery stress may be sensitive to the high strain rate of mortar, resulting in a decrease in its dynamic compressive strength.

## 5. Summary and Conclusions

This study investigated the dynamic compressive behavior of SMA fiber-reinforced mortar under a high strain rate. For this purpose, the plain specimen and specimens reinforced with 1.5% volume fiber of crimped and dog-bone-shaped fibers were prepared. The impact test was conducted using the split Hopkinson pressure bar method with impact velocities of 8 and 12 m/s. The following conclusions were drawn based on the findings of this study:

SMA fibers increase the strain rate of reinforced specimens because the fibers are more homogeneous than the mortar matrix. Moreover, the recovery stress of heated SMA fibers affects the transferred impact waves in the reinforced specimens, thus increasing the strain rate. The composite capacity of SMA fibers also influences the strain rate: the higher the composite capacity, the higher the strain rate.

The specimen changes the failure mode from longitudinal splitting failure at strain rates less than 17 s^−1^ to pulverization failure at a strain rates higher than 49 s^−1^. The cracks continuously propogate in the fragments of reinforced specimens because of the holding effect of SMA fibers, resulting in smaller fragments.

The dynamic compressive strength decreases with the addition of SMA fibers due to the porosity and overlap of the interfacial transition zone aureoles and the bridging effect of the SMA fibers. Heating damages the cement paste and evaporates the pore water, thus reducing the dynamic strength. The composite behavior between SMA fibers and the mortar matrix affects the dynamic compressive strength: samples with more composite have less dynamic compressive strength. The dog-bone-shaped fibers lose the anchoring bond due to heating; thus, they are poorly composited with the mortar matrix, leading to an increasing dynamic compressive strength of DG(H) specimens. In contrast, the increasing pullout resistance of the crimped fiber due to heating causes a decreasing dynamic compressive strength of CR(H) specimens.

The peak strain of reinforced specimens is higher than that of the plain specimen because the SMA fibers result in more cracks before the peak stress. Heating damages the mortar properties, leading to the appearance of more cracks and an increase of the peak strain. The dog-bone-shaped fiber with lower composite compacity than that of the crimped fiber produces a higher peak strain.

The specific energy absorption increases with increasing strain rate. The SMA fibers produce more specific energy absorptions because they bridge cracks, requiring more energy. However, the appearance of SMA fibers and the recovery stress do not affect the specific energy absorption very much.

At strain rates less than the transition strain rate of 30 s^−1^, the mechanical properties of the impacted specimens at a the velocity of 8 m/s can be predicted by linear functions of strain rate, increasing trends of dynamic compressive strength, and decreasing trends of the peak strain and specific energy absorption.

The elastic moduli of the plain and reinforced specimens, however, are not influenced by the strain rate but are affected by the correlation between elastic moduli of SMA fibers and the mortar matrix. The lower elastic modulus of the SMA fibers reduce the elastic modulus of the reinforced specimen.

Overall, the appearance of SMA fibers has a negative effect on the dynamic compressive properties of mortar; moreover, the effect of recovery stress is not clear. However, the composite behavior between the SMA fibers and the mortar matrix significantly influences the dynamic behavior. The dynamic compressive strength of the DG specimen is lower than that of the CR specimen, whereas peak strain shows the opposite trend because of the higher composite capacity of the crimped fibers.

## Figures and Tables

**Figure 1 materials-14-04933-f001:**
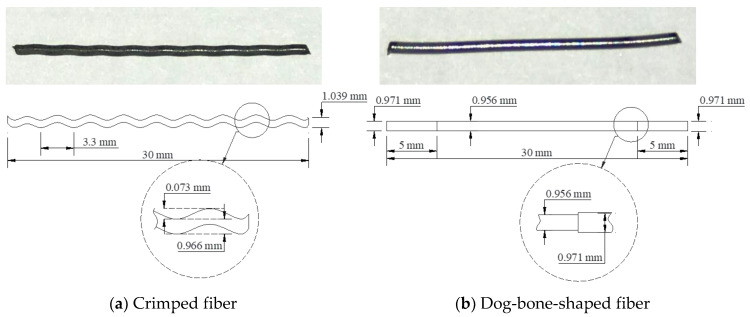
Dimensions of crimped and dog-bone-shaped fibers.

**Figure 2 materials-14-04933-f002:**
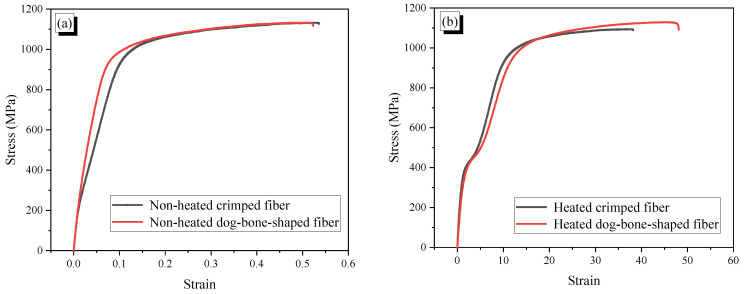
Tensile behavior of crimped and dog-bone-shaped fibers: (**a**) non-heated fibers; (**b**) heated fibers.

**Figure 3 materials-14-04933-f003:**
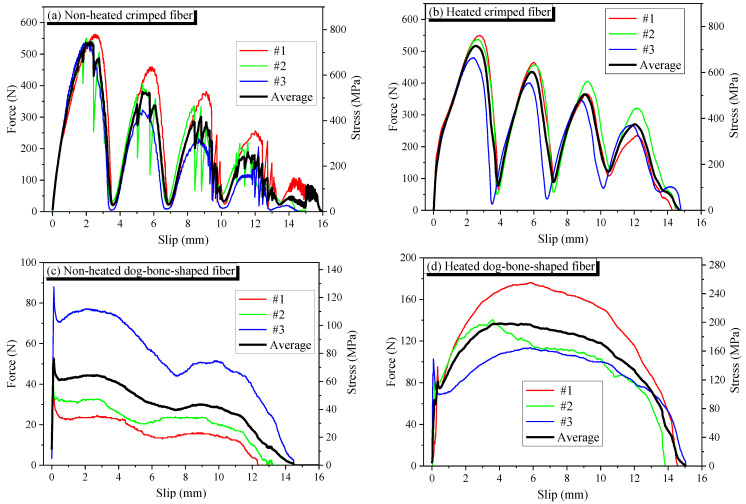
Pullout behavior of crimped and dog-bone-shaped fibers: (**a**) non-heated crimped fiber; (**b**) heated crimped fiber; (**c**) non-heated dog-bone-shaped fiber; (**d**) heated dog-bone-shaped fiber.

**Figure 4 materials-14-04933-f004:**
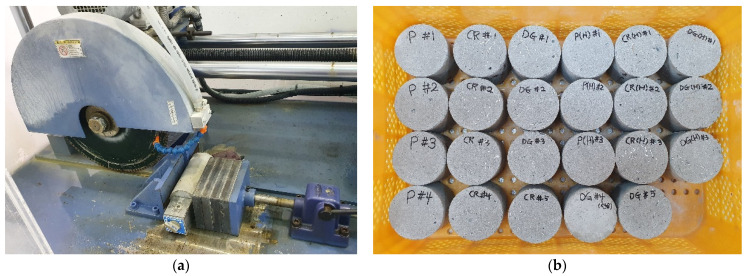
Cutting procedure and final D75 × H75 mm specimens: (**a**) cutting procedure; (**b**) final D75 × H75 mm specimens.

**Figure 5 materials-14-04933-f005:**
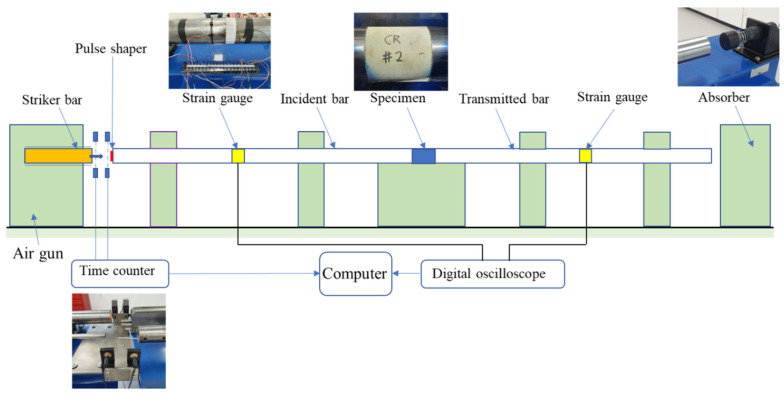
SHPB test set-up.

**Figure 6 materials-14-04933-f006:**
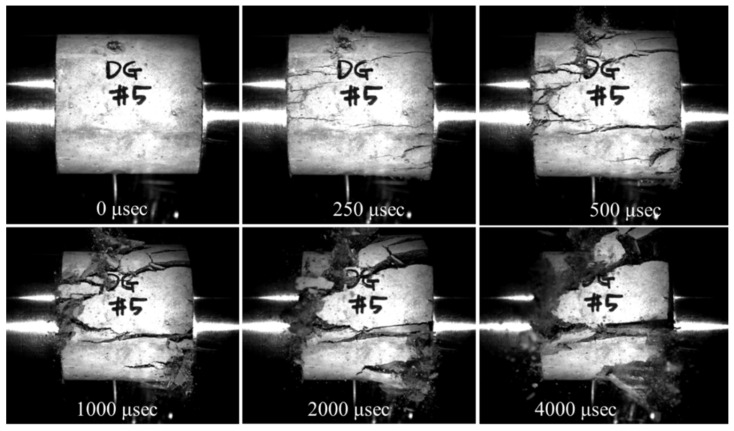
Images of the DG specimen in the SHPB test.

**Figure 7 materials-14-04933-f007:**
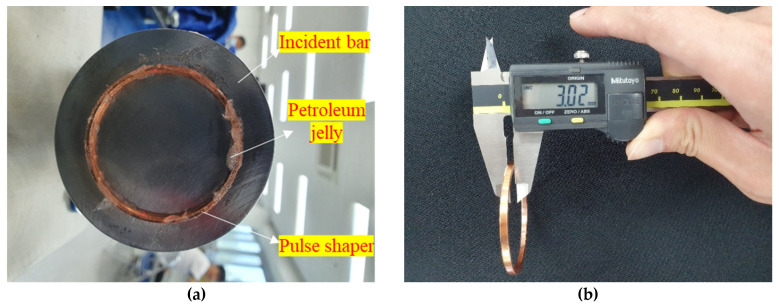
Attached annular pulse shaper and its thickness: (**a**) attached annular pulse shaper; (**b**) thickness of the annular pulse shaper.

**Figure 8 materials-14-04933-f008:**
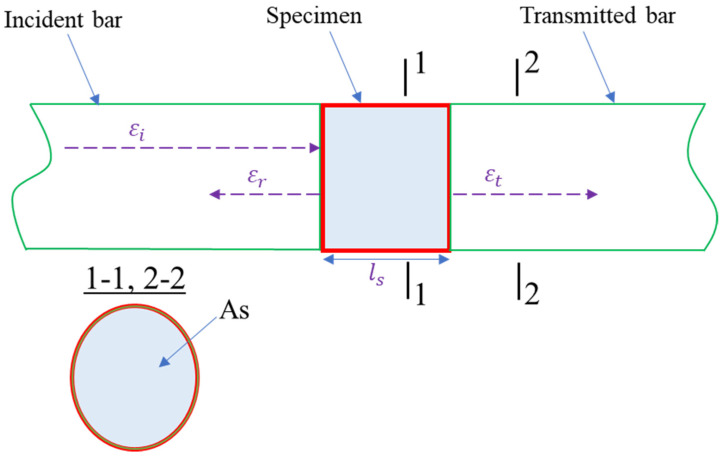
Schematic of the strain pulse.

**Figure 9 materials-14-04933-f009:**
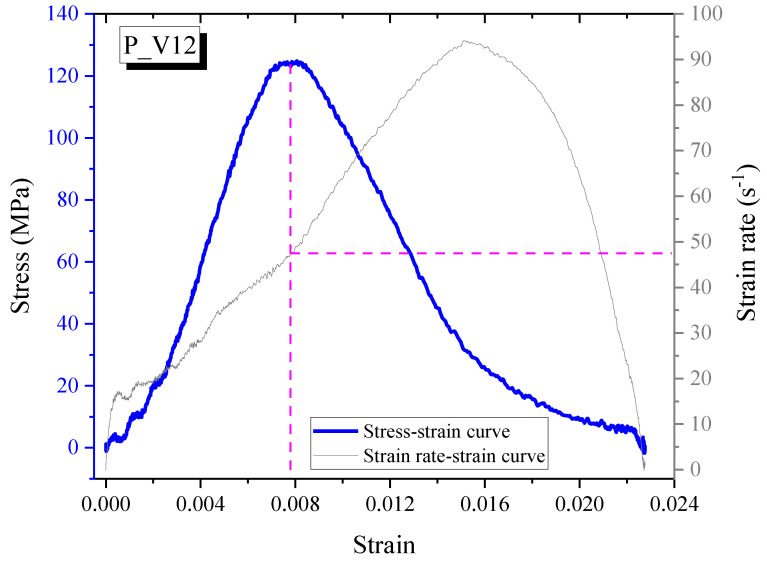
The curves of stress-strain and strain rate-strain at an impact velocity of 12 m/s.

**Figure 10 materials-14-04933-f010:**
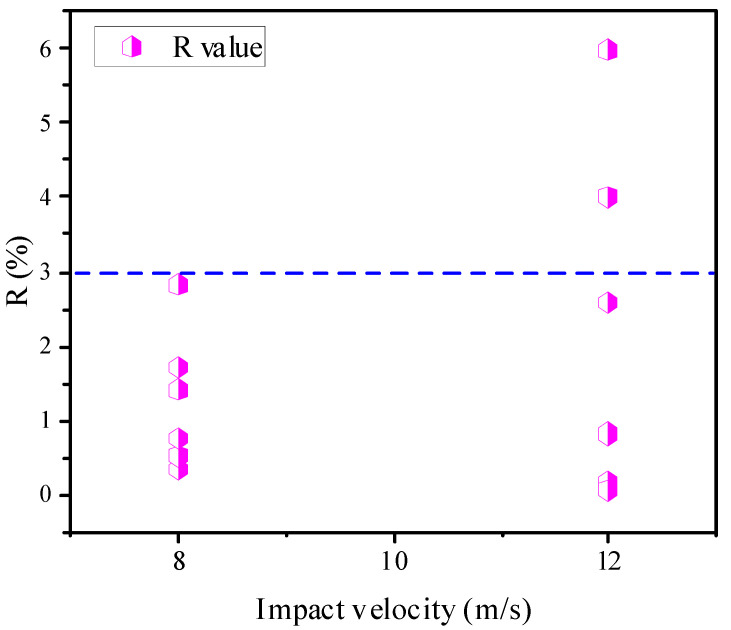
R-values of the specimens at impact velocities of 8 and 12 m/s.

**Figure 11 materials-14-04933-f011:**
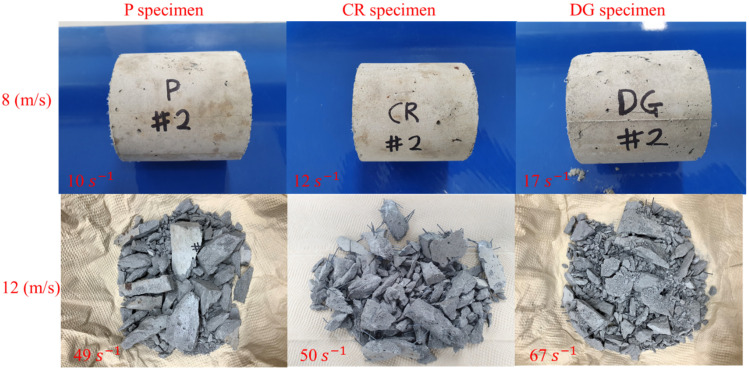
Fragments of non-heated specimens.

**Figure 12 materials-14-04933-f012:**
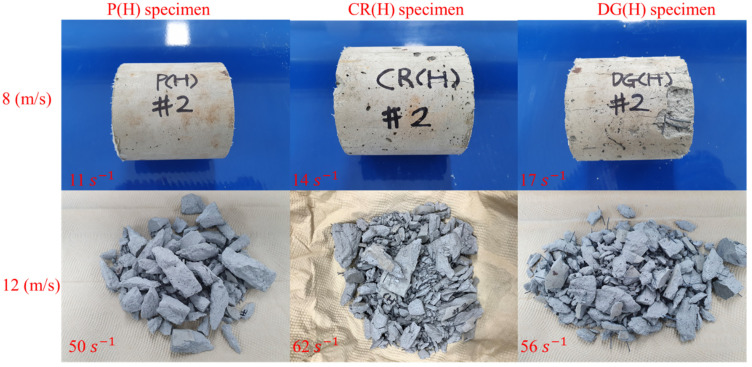
Fragments of heated specimens.

**Figure 13 materials-14-04933-f013:**
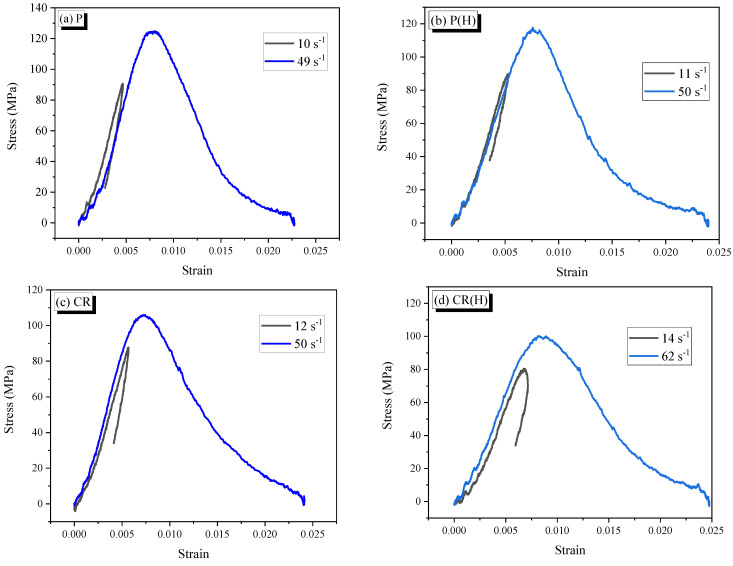
stress-strain curves of specimens at various strain rates.

**Figure 14 materials-14-04933-f014:**
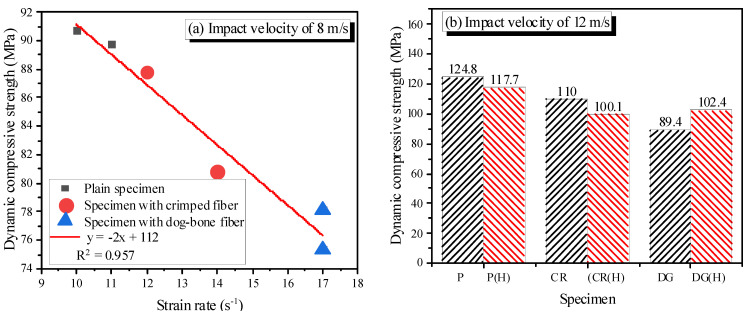
Dynamic compressive strength of the specimens: (**a**) impact velocity of 8 m/s; (**b**) impact velocity of 12 m/s.

**Figure 15 materials-14-04933-f015:**
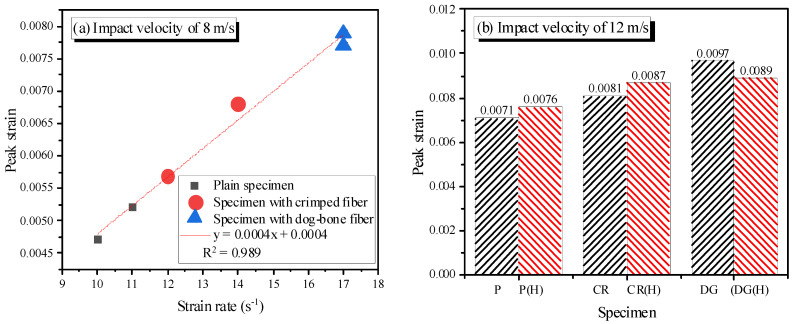
Peak strain of the specimens: (**a**) impact velocity of 8 m/s; (**b**) impact velocity of 12 m/s.

**Figure 16 materials-14-04933-f016:**
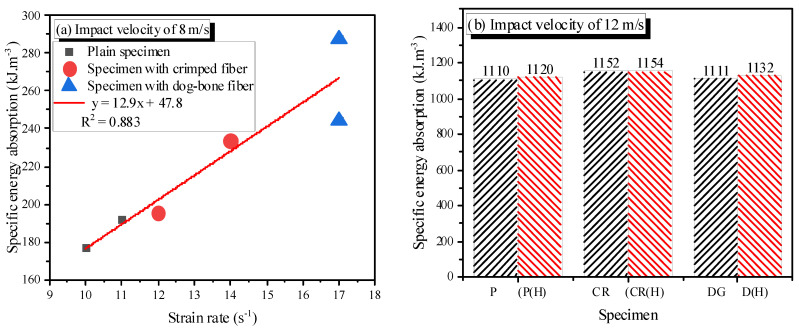
Specific energy absorption of the specimens: (**a**) impact velocity of 8 m/s; (**b**) impact velocity of 12 m/s.

**Figure 17 materials-14-04933-f017:**
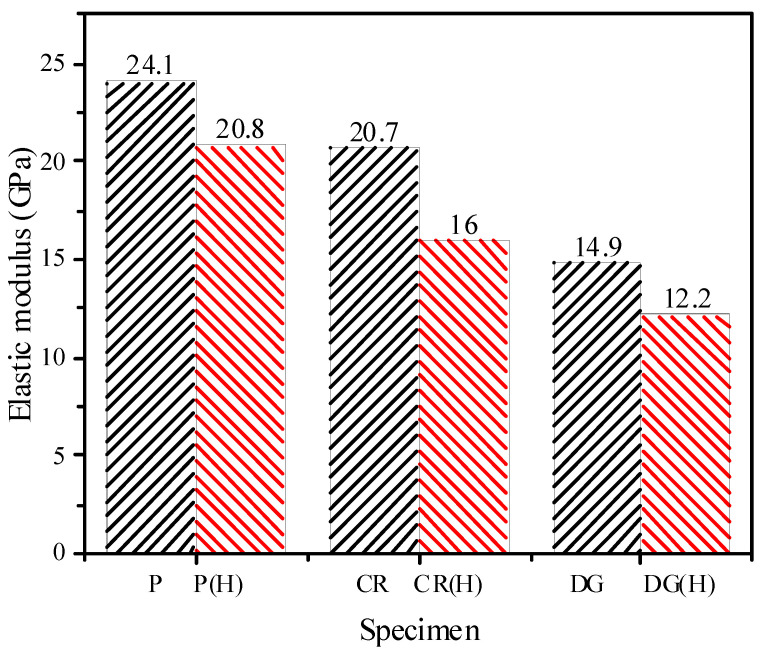
Elastic moduli of the specimens.

**Table 1 materials-14-04933-t001:** Composition of mortar with the weight ratios.

Cement (Type I)	Fly Ash	Silica Sand	Water-Reducing Admixture	Water
1.00	0.15	1.00	0.009	0.35

**Table 2 materials-14-04933-t002:** Front, back, and average stresses and R-values of the specimens.

Num.	Specimen	Front Stress (MPa)	Back Stress (MPa)	Average Stress (MPa)	R-Value (%)	Strain Rate (s^−1^)
1	P_V8	90.8	90.5	90.7	0.4	10
2	CR_V8	88.4	87.2	87.8	1.4	12
3	DG_V8	79.2	77.0	78.1	2.8	17
4	P(H)_V8	90.0	89.5	89.7	0.5	11
5	CR(H)_V8	81.5	80.1	80.8	1.7	14
6	DG(H)_V8	75.5	75.0	75.3	0.8	17
7	P_V12	124.7	124.9	124.8	0.2	49
8	CR_V12	110.0	109.9	110.0	0.1	50
9	DG_V12	92.1	86.7	89.4	5.9	67
10	P(H)_V12	120.1	115.4	117.7	4.0	50
11	CR(H)_V12	101.4	98.8	100.1	2.6	62
12	DG(H)_V12	102.8	102.0	102.4	0.8	56

**Table 3 materials-14-04933-t003:** Mechanical properties of specimens under high strain rates.

Specimens	Dynamic Compressive Strength (MPa)	Peak Strain (×10^−3^)	Specific Energy Absorption (kJ·m^−3^)
P_V8	90.7	4.7	177
CR_V8	87.8	5.7	196
DG_V8	78.1	7.7	245
P(H)_V8	89.7	5.2	192
CR(H)_V8	80.8	6.8	234
DG(H)_V8	75.3	7.9	288
P_V12	124.8	7.1	1110
CR_V12	110.0	8.1	1152
DG_V12	89.4	9.7	1111
P(H)_V12	117.7	7.6	1120
CR(H)_V12	100.1	8.7	1154
DG(H)_V12	102.4	8.9	1132

## Data Availability

Not applicable.

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
