# Peer review of "Dynamic Behaviors of Mortar Reinforced with NiTi SMA Fibers under Impact Compressive Loading"

_materials, 2021, doi:10.3390/ma14174933_

Round 1

Reviewer 1 Report

This manuscript investigates the dynamic behaviour of mortars reinforced with SMA fiber under different impact loading by using SHPB. The dynamic stress strain curve, compressive strength and elastic modulus of mortars containing Crimped and Dog bone shaped fiber in mortar samples were compared and discussed. Indeed, the test results provided some new information to the related field, and the manuscript is well constructed. However, some information still needs to be added, and questions are remaining to be clarified. The comments and suggestions are as follows:

  1. In the part of Introduction, the research gaps are still not clear, and the motivation of this study is not well illustrated. The authors said this is the final puzzle piece in the compressive behaviour of SMA fiber reinforced mortars, but why?
  2. In section 2.1, the detailed information of sand, fly ash and water reducer should be added, for example, particle size distribution, type of SP. It is better to give the mixture design based on 1m3 of mortar.
  3. In Table 1, it is water reducing admixture, not water reducin admixture.
  4. Line 120, D75×H150, the unit is mm? Please use the correct formation.
  5. Line 123, actually, 150°C is such a high temperature for cement based materials, part of C-S-H and Aft(or AFm) will lose the bound water after heating, consequently, the shrinkage induced cracks may present and influence the mechanical behaviour of concrete. Therefore, how did the authors evaluate the effects of pre-heating on the dynamic performance of samples? Does that mean the SMA is not suitable for concrete if pre-heating is necessary?
  6. Line 150, the authors select 8 and 12 m/s as the impact velocities, these values are according to any regulations or test standards? Please specify it.
  7. Line 219, the R value of tested samples ranges from 0-6%, the authors believe they are reliable, so what is the critical value for R?
  8. In Table 3, the specific energy absorptions are presented, please provide the detailed information of calculation.
  9. Line 362, moduli should be modulus.
  10. Figure 13, it is necessary to explain why the stress of DG specimens after heating was higher than that of unheated group, which was different from P and CR specimens?
  11. Figure 17, Please explain in detail the essential reason why heating reduces the elastic modulus.
  12. An overall discussion combining all collected results is necessary for this manuscript to explain the detailed kinetics of applying SMA fiber in mortars under dynamic loading, which was missed in this manuscript.

Author Response

Dear Reviewers:

The authors appreciate for your reviews, and we revised the paper following your comments.

Sincerely yours,

Reviewer 2 Report

The effects of the crimped fibers and dog-bone-shaped fibers on the compressive strength with dynamic behaviors. This is a very well-written paper. Below are just some minor comments and suggestions for improvement. Some suggestions for the current manuscript are as follows:

- The literature review merely lists the published work and does not present the major findings from each effort and how it ties to the presented research. The introduction lists some relevant research but fails to present a scientific review. Please consider to rewrite and clarify the motivation, and significance of this study.

- The objective statements are rather vague and lacks projected outcomes or how the paper will assist practitioners.

- Some necessary descriptions of the test results should be added, and more in-depth analysis should be added.

- The aspects of novelty of the paper must be pointed out more clearly due to the importance and complexity of the subject addressed in the work, particularly in the abstract and concluding remarks sections

- The authors should rewrite the abstract to include a summary of the key conclusions, clearly state the purpose of the work, the scope of the effort, the procedures used to execute the work, and major findings.

- Change "Conclusions" to "Summary and Conclusions", and rewrite the conclusions into specific conclusions.

Author Response

(The authors gave the same response as above.)

Round 2

Reviewer 1 Report

The authors have detailedly revised the paper according to the comments and the quality of the paper has been significantly improved.